# Facile Electrochemical Determination of Methotrexate (MTX) Using Glassy Carbon Electrode-Modified with Electronically Disordered NiO Nanostructures

**DOI:** 10.3390/nano11051266

**Published:** 2021-05-12

**Authors:** Aftab A. Khand, Saeed A. Lakho, Aneela Tahira, Mohd Ubaidullah, Asma A. Alothman, Khoulwod Aljadoa, Ayman Nafady, Zafar H. Ibupoto

**Affiliations:** 1School of Life Sciences, Tsinghua University, Beijing 100084, China; 2Department of Physiology, University of Sindh, Jamshoro 76080, Sindh, Pakistan; 3Department of Pharmaceutical Chemistry, Faculty of Pharmacy and Pharmaceutical Sciences, University of Karachi, Karachi 75270, Sindh, Pakistan; 4Dr. M.A Kazi Institute of Chemistry, University of Sindh, Jamshoro 76080, Sindh, Pakistan; aneelatahira80@gmail.com; 5Department of Chemistry, College of Science, King Saud University, Riyadh 11451, Saudi Arabia; mtayyab@ksu.edu.sa (M.U.); aalsalme@ksu.ed.sa (A.A.A.); 437204364@student.ksu.edu.sa (K.A.); anafady@ksu.edu.sa (A.N.)

**Keywords:** methotrexate, NiO nanostructures, sodium sulfate, electrochemical determination

## Abstract

Recently, the oxidative behavior of methotrexate (MTX) anticancer drug is highly demanded, due to its side effects on healthy cells, despite being a very challenging task. In this study, we have prepared porous NiO material using sodium sulfate as an electronic disorder reagent by hydrothermal method and found it highly sensitive and selective for the oxidation of MTX. The synthesized NiO nanostructures were characterized by scanning electron microscope (SEM) and X-ray diffraction (XRD) techniques. These physical characterizations delineated the porous morphology and cubic crystalline phase of NiO. Different electrochemical approaches have been utilized to determine the MTX concentrations in 0.04 M Britton–Robinson buffer (BRB) at pH 2 using glassy carbon electrode (GCE)-modified with electronically disordered NiO nanostructures. The linear range for MTX using cyclic voltammetry (CV) was found to be from 5 to 30 nM, and the limit of detection (LOD) and limit of quantification (LOQ) were 1.46 nM and 4.86 nM, respectively, whereas the linear range obtained via linear sweep voltammetry (LSV) was estimated as 15–90 nM with LOD and LOQ of 0.819 nM and 2.713 nM, respectively. Additionally, amperometric studies revealed a linear range from 10 to70 nM with LOD and LOQ of 0.1 nM and 1.3 nM, respectively. Importantly, MTX was successfully monitored in pharmaceutical products using the standard recovery method. Thus, the proposed approach for the synthesis of active metal oxide materials could be sued for the determination of other anticancer drugs in real samples and other biomedical applications.

## 1. Introduction

Methotrexate (MTX), is prescribed by doctors as an anticancer drug. MTX is chemically called 2,4-diamino-N-10-methyl folic acid, and it is prescribed for the treatment of various carcinomas such as acute lymphoblastic leukemia [1,2], head and neck cancer, gastric cancer, breast cancer, choriocarcinoma, etc. (www.druginfosys.com, accessed on 20 January 2017). After oral administration of MTX, it absorbs in 1–2 h inside the body. It has a mean oral bioavailability of 60% at low doses [3,4]. It has been found that MTX possesses significant toxicity that stopped the development of healthy cells, and thus, its clinical recommendations are restricted [5,6]. Hence, fast and sensitive methodologies should be developed to detect MTX in order to avoid the side effects offered by MTX. Furthermore, MTX possesses a low safety window; thus, it is necessary to monitor its therapeutic levels in the human body. Various analytical methods have been developed for the monitoring of MTX, such as enzyme multiplied immunoassay, radioimmunoassay, enzyme inhibition, and protein binding, fluorimeter, microbiological, or high performance liquid chromatography (HPLC) assays [7,8,9,10,11,12,13,14]. For measurement of MTX from blood samples, HPLC is the most reliable and predominantly used technique [15,16,17,18,19]. These methods are restricted due to their high cost and complexity of the operation. Besides these classical methods, electrochemical methods are also developed for the analysis of MTX [20,21,22,23,24,25,26,27,28,29,30,31,32,33,34,35,36]. Electroanalytical methods are low cost, simple, selective, sensitive, very fast in response, and easy-to-handle, on-spot measurements. These methods require highly active electrocatalytic materials that can measure MTX selectively under in vitro and real sample conditions. This is now possible due to the manipulation of materials at the atomic level because of revolutionary advancements in the field of nanotechnology. For this purpose, intensive efforts are used to fabricate earth-abundant and low-cost metal oxide nanostructures such as NiO, CuO, TiO_2_, ZnO, and their hybrid structures with enhanced electrochemical activities [35,37,38,39,40,41,42,43]. Nanostructured materials offer a high surface area for the sensing applications; thus, different morphologies at the nano level have shown outperforming properties in sensing electronics and optoelectronics applications [44,45,46]. The unique properties of nanostructures in a specific application can be tuned by alteration in the size or shape. For the electroanalytical method, the surface of the material is a key factor that governs the sensitivity, selectivity, stability, and fastness of the analytical method. NiO is among these materials, which has shown excellent performance in catalytic applications due to its fascinating surface area [47,48,49]. Still, more work is needed to tailor the electronic and surface properties of NiO for in vitro analysis and possible commercial applications. There is no report for the use of sodium sulfate as a template to tune the electronic and surface properties of NiO. The sulfate ion carries a double-negative charge, which can actively bind with nickel ions, and consequently, a complex may be formed during the growth. Thus, the competition of sulfate ions and hydroxide in the growth process could alter the electronic structure and surface properties of NiO due to calcination at 500 °C in air, which can result in an impurity-free NiO. The produced nanostructures of NiO have shown robust catalytic activities toward the oxidation of MTX in the BRB buffer solution of pH 2.

In this study, we propose a facile, fast, easy, and sensitive method for the analysis of MTX using NiO porous material. NiO nanostructures were produced by hydrothermal method using sodium sulfate as an electronic disorder reagent. The NiO nanostructures are porous in morphology and exhibited cubic crystallography. Further, these nanostructures were used for the sensing of MTX in 0.01 M Britton–Robinson buffer (BRB) solution of pH 2. The MTX sensor exhibits a wide linear range from 15 to 90 nM and low LOD and LOQ 0.1 nM, and 1.3 nM, respectively, using the LSV mode of analysis.

## 2. Material and Methods

### 2.1. Chemical and Reagent

All reagents used were of analytical grade. Methotrexate (MTX) in table form was purchased from the local market of Karachi, Sindh, Pakistan. Phosphoric acid was purchased from Biom Laboratories Ltd. (Neu-Isenburg, Germany). Nickel chloride, sodium sulfate, urea, boric acid, and sodium hydroxide (NaOH), were purchased from Sigma-Aldrich, Karachi, Pakistan. Hydrochloric acid was purchased from Merck. The stock solution of 1 mM MTX was prepared in 0.04 M BRB buffer solution of pH 2. All the desired solutions were prepared in deionized water.

### 2.2. Synthesis of NiO Nanostructures

NiO nanostructures were obtained through the hydrothermal method, and the used methodology is described as follows: The nickel chloride hexahydrate of 0.1 M concentration was mixed with 0.1 M urea in two separate beakers, and an amount of 50 and 100 mg of sodium sulfate as electronic disorder reagent was also added. The purpose of using different quantities of sodium sulfate was to evaluate the role of sulfate ions on the catalytic and structural properties of NiO. Both beakers were named sample 1 and sample 2. The growth solutions were covered with an aluminum sheet and thermally treated at 90 °C for 5 h in an electric oven. The obtained nanostructured samples were collected on the filter paper and dried overnight. Then, calcination at 500 °C was carried out on these samples for 4 h in air to obtain NiO nanostructures. We repeated the experiment more than three times and we still observed the same cubic phase and morphology of NiO. A similar method was used to prepare a pure NiO sample without the use of sodium sulfate. The top view of NiO nanostructures was observed through SEM at 3 kV. The crystal arrays of NiO nanostructures were studied by powder X-ray diffraction (XRD) using experimental conditions of CuKα radiation (λ = 1.54050 Å), 45 mA, and 45 kV.

### 2.3. Electrochemical Sensing of MTX Anticancer Drug Using NiO Nanostructures

Potentiostat Ametek Versa STAT 4 (Tilburg, the Netherlands) was used to evaluate the electrochemical properties of NiO using a three-electrode configuration. The electrochemical cell consisted of the following electrodes: silver-silver chloride (Ag/AgCl) as reference electrode, platinum wire as the counter electrode, and the NiO nanostructure-modified GCE as working electrode. The CV, LSV, and amperometric modes were adapted to characterize the prepared NiO nanostructures for the sensing of MTX in a 0.04 M BRB buffer solution of pH 2. A catalyst ink was made with 5 mg of NiO in 1 mL of deionized water at a constant sonication process for 30 min for obtaining homogeneous slurry. The 5% Nafion with a volume of 20 μL was added as a binder and adherent material for the NiO on GCE. The cleaning of GCE was performed with alumina paste and a silicon sheet. A 10 μL of NiO ink was deposited on GCE using a drop-casting strategy and dried at ordinary conditions. The high concentration solution of MTX was made in 0.04 M BRB of pH 2 using 10 mg of MTX tablet. The concentration of MTX solution was 1 mM, and low concentrations were prepared from this solution using the dilution method. The electrochemical experiments were performed at standard conditions.

## 3. Results and Discussion

### 3.1. Morphology and Crystallography Studies of Prepared NiO Nanostructures

Figure 1A illustrates the shape structure of nanostructured NiO obtained with and without sodium sulfate as an electronic disorder reagent. The pristine NiO is well occupied by the flower-like structure, as shown in Figure 1A(a). The dimension of each flower was approximately 500 nm. However, the addition of sodium sulfate resulted in porous features of NiO (sample 1 and sample 2), and the average thickness of walls of each pore was 150 nm to 300 nm as shown in Figure 1A(b,c). From the top surface analysis, it is clear that using a sodium sulfate could enlarge the structure with a more defined morphology, which has shown a great impact on the functional properties of NiO (sample 2). In the growth process, sulfate ions from the sodium sulfate could be complexed with Ni ions due to their heavy charge, compared to hydroxide ions; thus, they have evolved the porous morphology due to the calcination at 500 °C in air, which prevented impurity of sulfur in the final product of NiO. The crystal arrays of pristine NiO, sample 1, and sample 2 were investigated by powder XRD, as enclosed in Figure 1B. The pristine NiO experienced a well-judged cubic phase geometry, and the measured crystal planes are in good agreement with reference card no. 01-089-7130, as shown in Figure 1Ba. Sample 1 and sample 2 were also studied by XRD, as shown in Figure 1B(b,c), and the measured reflections confirm the cubic phase of NiO. Additionally, they carry an intense 002 peak, which is the characteristic peak of NiO, and the measured results of XRD are in excellent consistency with reference card no. 01-089-7130.

### 3.2. Electrochemical Oxidation of MTX

The electrochemical characterization on pristine NiO, bare GCE, sample 1, and sample 2 was evaluated by CV at 50 mV/s in 0.04 M BRB buffer solution of pH 2, as shown in Figure 2a. All the materials did not show any redox reaction in the electrolytic solution, which implies the negligible electrochemical activity of them. However, CV was also performed on these materials in the presence of 20 nM MTX prepared in 0.04 M BRB buffer solution of pH 2, as shown in Figure 2b. The bare GCE, pristine NiO, and sample 1 did not show any electrochemical activity toward the oxidation of MTX. Figure 2a,b shows different volumes of solution in the electrochemical cell since Figure 2a indicates the response of each material in the electrolytic conditions, whereas Figure 2b shows the response of each material with electrolyte and the MTX as analyte; therefore, this might influence on the value of current, and we observe similar CV features but with different background of current for the S1. However, sample 2 has demonstrated a well-resolved oxidation peak current at 0.95 V with an enhanced current, which testified to the use of sodium sulfate as an electronic disorder reagent for the NiO nanostructures. Further, sample 2 shows a superior sensitivity over other materials, and thus, it was used for the rest of the electrochemical characterization toward the sensing of MTX. The superior performance of the sample is attributed to several aspects of nanostructured material such as the particle size, porosity of the material, and surface roughness since they provide a large number of catalytic sites for the oxidation of anticancer drugs. Surface roughness and porosity provide the exposure of a large surface area for the catalytic reaction, and thus, superior performance is shown by sample 2 in the reported work.

From CV data, it can be inferred that the oxidation of MTX while considering an anodic peak is irreversible and associated with the electro-oxidation of the pyrazine species through the mechanism of two electrons and two protons process [50,51]. In the current study, NiO is providing a large surface area to the MTX, which further facilitates the oxidation of MTX, actively capturing the electrons ejected by the MTX. Therefore, favorable oxidation of MTX was detected. The scan rate study was performed on sample 2 at different scan rates of 10, 15, 20, 40, 60, and 80 mV/s for the description of electrode kinetics in 20 nM of MTX, as shown in Figure 2c. The oxidation peak current was increased with each increment of scan rate, which revealed a direct electron transfer between the modified electrode and electrolyte. A plot of peak current versus scan rate shows an excellent linear fit with a regression coefficient of 0.99, suggesting a diffusion-controlled oxidation reaction of MTX on sample 2, as shown in Figure 2d. The CV was employed to record the linear range of MTX based on sample 2, as shown in Figure 3a. The increase in the concentration of MTX brought a successive increase in the oxidation peak current. The peak current was plotted versus different concentrations of MTX, as shown in Figure 3b. From the linear fit, the working range for the MTX was observed from 5 to 30 nM with LOD and LOQ as 1.459 nM and 4.86 nM, respectively. This study indicates that a sensor has a high potential to demonstrate the determination of MTX from a wide range of real samples. Chronoamperometry was used to record the calibration plot and to observe the sensitivity of sample 2 toward oxidation of MTX. Therefore, the i–t curve was collected at 0.95 V with an injection of MTX concentrations, as shown in Figure 3c. It is clear from the obtained results that current was increasing with each increment of MTX concentration, indicating the effectiveness of sample 2 toward the oxidation of MTX in 0.04 M BRB buffer of pH 2. The measured current was plotted against different concentrations of MTX, as shown in Figure 3d. The best linear fit was obtained for the concentration range from 10 to 70 nM with a regression coefficient of 0.99, which confirms the robust analytical performance of sample 2. Additionally, the LOD of 0.1 nM was calculated from amperometry data. The LSV was used at the scan rate of 50 mV/s for the development of the calibration plot for MTX, as shown in Figure 4a. From LSV, it can be seen that the oxidation peak current was linearly increasing with the addition of MTX concentration, suggesting the superfast sensitivity of sample 2 toward the oxidation of MTX. The peak current from LSV was plotted versus different concentrations of MTX, as shown in Figure 4b. The well-deserved fitting is indicating excellent analytical features of sample 2 for the quantification of MTX. The linear range for the MTX was found to be 15–90 nM. The LOD and LOQ were determined as 0.819 nM and 2.713 nM, respectively. The possible reasons for the different performance by CV and LSV might be explained by noting that CV uses both forward and reverse processes that may change the electrochemical properties of NiO, and it could decrease the activity of catalytic material during the several cycles of CV. Therefore, we observed the different sensitivity toward the detection of MTX using CV and LSV. Additionally, we did not optimize electrochemical cell conditions such as the optimum electrolyte volume, the position of the working electrode inside the cell, and the catalytic material on the glassy carbon electrode. All these conditions should be controlled during the analysis of practical analysis of drugs or any other analyte. However, we investigated the electrochemical properties of NiO for MTX, and herein report the best result that we obtained.

The selectivity of the MTX device based on sample 2 was examined in the presence of 40 nM MTX using CV and other interfering substances, as shown in Table 1. The use of 1:10 ratio of MTX and interfering substances volumes of similar concentration did not adhere to any possibility of interference. The interfering substances were selected on the basis of complexity and their presence in the blood samples. These include 5-fluorouracil, mitoxantrone, glucose, sucrose, urea, and Cl^−^. The tolerance limit was calculated according to the reported work [52], in which the tolerance limit of each interfering species produced an interference of less than 5%. These estimations are made on the basis of change in the peak current of MTX when an interfering species was added. Table 1 shows the obtained tolerance limit for the common interfering species, confirming an excellent selectivity of the proposed material (sample 2) toward MTX determination. The NiO is stable at standard conditions and we observed significant stability of the proposed NiO-modified GCE for the period of 3 weeks through the CV. Additionally, the electrode has excellent repeatability performance when observed in the 10 cycles of CV in 20 nM MTX. The real sample analysis capability of sample 2 was investigated through the recovery method with close values to that of MTX in blood samples. These recovery values are given in Table 2. The excellent percent recovery justified the high effectiveness of the proposed analytical method for the determination of MTX from blood and pharmaceutical samples. The performance of the proposed analytical method was compared with recently reported methods, as given in Table 3 [24,31,53,54]. It is clear that the proposed method has several advantages such as simple materials synthesis, excellent linear range, and limit of detection, and thus, it can be regarded as a potential method for the monitoring of MTX from real samples.

## 4. Conclusions

In summary, we have used sodium sulfate as an electronic disorder reagent in the synthesis of NiO nanostructures using the hydrothermal method. A porous structure was evident for the as-prepared NiO with a cubic unit cell. In particular, sample 2 of the prepared NiO nanostructures was highly active toward catalytic oxidation of MTX at pH 2 of BRB buffer solution. The fabricated MTX sensor exhibited a linear range from 10 to 70 nM with low LOD and LOQ of 0.1 nM and 1.3 nM, respectively, using amperometry mode. Given that the developed strategy for the synthesis and fabrication of NiO/GCE is inexpensive, simple, and environmentally friendly, it could pave the way for large-scale production of MTX sensors for monitoring of MTX in pharmaceutical and blood samples. Significantly, the use of sodium sulfate as an electronic disorder reagent in a wide range of metal oxides will enable their utilization in an extended range of biomedical and energy applications.

## Figures and Tables

**Figure 1 nanomaterials-11-01266-f001:**
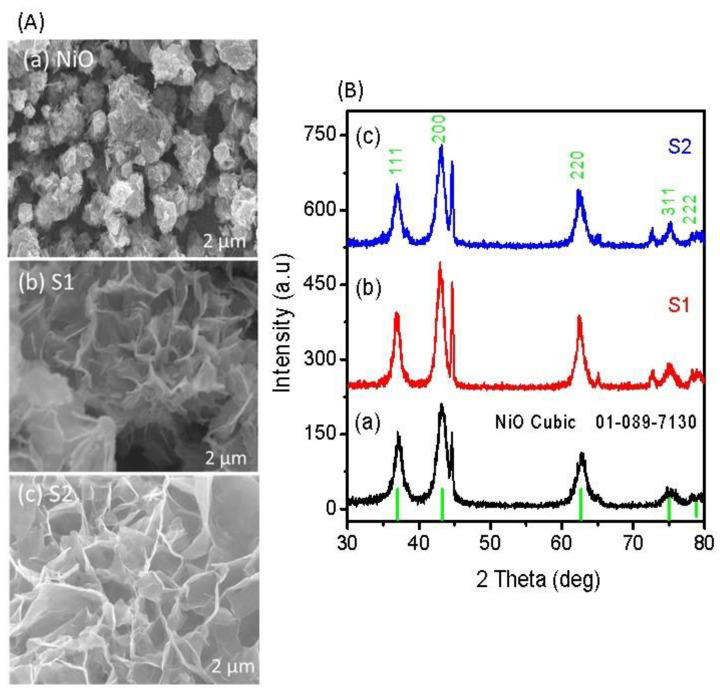
(**A**) SEM images of (**a**) pristine NiO, (**b**) sample 1, and (**c**) sample 2; (**B**) XRD diffraction patterns of (**a**) pristine NiO, (**b**) sample 1, and (**c**) sample 2.

**Figure 2 nanomaterials-11-01266-f002:**
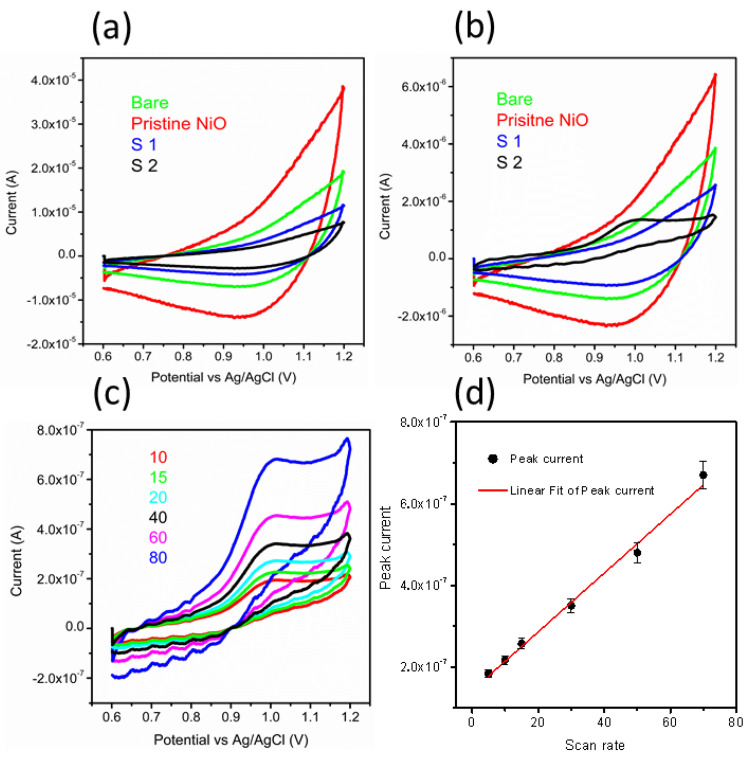
(**a**) CV curves of bare GCE, pristine NiO, sample 1, and sample 2 in an electrolytic solution of 0.04 M BRB buffer of pH 2 at a sweeping rate of 50 mV/s; (**b**) CV curves of bare GCE, pristine NiO, sample 1, and sample 2 at a sweeping rate of 50 mV/s in 20 nM MTX prepared in a 0.04 M BRB buffer of pH 2; (**c**) CV curves recorded for sample 2 at different scan rates of 15, 15, 20, 40, 60, and 80 mV/s in 20 nM MTX; and (**d**) A fitting of peak current versus different scan rates.

**Figure 3 nanomaterials-11-01266-f003:**
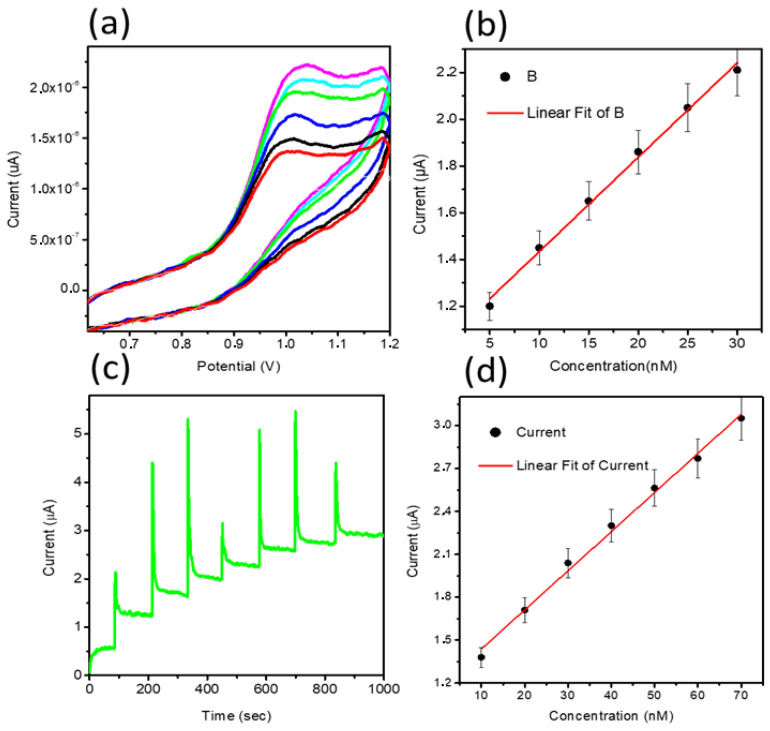
(**a**) CV curves recorded at a sweeping rate of 50 mV/s for sample 2 at various concentrations of MTX ranging from 5 to 30 nM in a 0.04 M BRB buffer of pH 2, (**b**) A plot of peak current against different concentrations of MTX from 5 to 30 nM, (**c**) i–t curve at the oxidation potential of 0.95 V MTX with the addition of 10 nM MTX, and (**d**) a fitting of increasing current versus various concentrations of MTX.

**Figure 4 nanomaterials-11-01266-f004:**
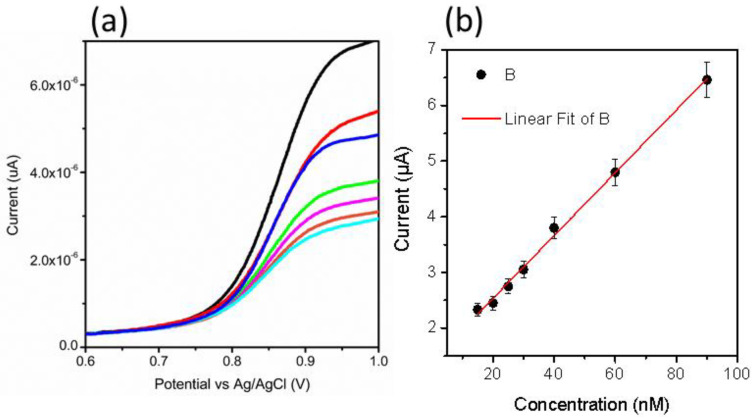
(**a**) LSV curves measured at 50 mV/s for different concentrations 15–90 nM of MTX in a 0.04 M BRB buffer of pH 2 and (**b**) linear plot of oxidation peak current versus different concentrations of MTX.

**Table 1 nanomaterials-11-01266-t001:** Selectivity results of sample 2 for MTX under the environment of common interfering species.

Interfering Reagent	Concentration (nM) ^a^	Signal Increase ^b^
5-Flourouracil	40	0.54
Mitoxantrone	40	1.32
Glucose	40	2.2
Sucrose	40	1.9
Urea	40	−2.3
Chloride	40	−0.65

^a^ Spiked concentration of 40 nM MTX in 25 mL. ^b^ Percent increase of analytical signal during the use of CV, followed by the addition of interfering molecules.

**Table 2 nanomaterials-11-01266-t002:** The percent recovery results of sample 2 during the determination of MTX.

Tablet Number	Added nM	Found nM	% Recovery	RSD (%)
1	10	9.98	99.8	1.23
	20	20.02	100.1	1.56
	30	29.99	99.96	0.98
2	10	10.01	100.2	1.45
	20	20.04	100.4	0.59
	30	29.97	99.9	1.59

**Table 3 nanomaterials-11-01266-t003:** The comparison of the figure of merits of the proposed analytical method with reported methods.

Electrode	Method	Linear Range	Low Detection Limit (nM)	Reference
MWCNT–SPE	SWV	0.5–100 µM	100	[18]
NanoCu/Carbon black	SWV	2.2–25	900	[19]
CoFe_2_O_4_/rGO/IL/GCE	DPV	0.05–7.5	10	[20]
CD-GNs/GCE	DPV	0.1–1.0	20	[21]
NiONS/GCE	CV	5–30 nM	1.459 nM	This work
	LSV	15–90 nM	0.819 nM	
	Amperometry	10–70 nM	0.1 nM	

## Data Availability

Data sharing not applicable.

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
