# Peer review of "Facile Electrochemical Determination of Methotrexate (MTX) Using Glassy Carbon Electrode-Modified with Electronically Disordered NiO Nanostructures"

_nanomaterials, 2021, doi:10.3390/nano11051266_

Round 1

Reviewer 1 Report

This manuscript describes a synthetic way to obtain a new electroactive material for anticancer drugs quantification. I think that this manuscript is worth of publication after minimal changes:

- The authors should add an uncertainties analysis to the results both on graphs and on the results. How many times the experiments were performed?

- On figure 2b units should be added

-The value of the oxidation potential should be indicated.

-A statements on why only two quantities of sodium sulphate was used. Have they obtained results with different quantities of sodium sulphate, and how is the reproducibility? 

Author Response

Reviewer 1

We are thankful to the reviewer for useful comments and suggestions prior to publication

Comments and Suggestions for Authors

This manuscript describes a synthetic way to obtain a new electroactive material for anticancer drugs quantification. I think that this manuscript is worth of publication after minimal changes:

- The authors should add an uncertainties analysis to the results both on graphs and on the results. How many times the experiments were performed?

Ans.  We acknowledge the reviewer comment and we report the best of electrochemical properties of the NiO nanostructures for MTX drug. However, we only repeated the synthesis process and physical characterization more than three times in order to check the structure, and purity of materials. As we used sulfate as disordering reagent and we could expect the impurity of sulfur in the prepared samples but we noticed all the times we have pure phase of NiO. These corrections related to uncertainties are added to the linear fitting graphs in the revised version of manuscript.

- On figure 2b units should be added

Ans. The unit has been added to the Figure 2b

-The value of the oxidation potential should be indicated.

Ans. This has been added in the revised version of manuscript.

-A statements on why only two quantities of sodium sulphate was used. Have they obtained results with different quantities of sodium sulphate, and how is the reproducibility? 

Ans. The purpose of using different quantities of sodium sulfate was to evaluate the role of sulfate ions on the catalytic and structural properties of NiO and this has been discussed in the paper. We have repeated the experiment for more than three times and we still observed the same electrochemical properties as described in the manuscript. We only reported the best functionalities of NiO for the detection of anticancer drug.

Reviewer 2 Report

This paper describes about NiO nanostructures for detecting methotrexate (MTX). I do believe this material is to be of interest and important. However, I don’t think this paper demonstrate the sufficiently scientific evidences and discussion to support the advantages of the developed MATERIAL. So I fail to see the quality and advantage of this study. Therefore, it is recommended that the authors should reconsider this paper (major revision).

Some comments are listed below.

Title

The title is included “new protocol for monitoring”. The proposed protocols are traditional electrochemical techniques (CV, LSV, and i-t). The authors should revise this part.

The developed NiO nanostructures

I understand that the obtained sample 2 exhibited superior electroanalytical performance to pristine NiO and sample 1. But it is unclear this reason. Due to the size of porous structure (surface roughness, or porosity)? The authors should explain this reason sufficiently because this point is the most important in this manuscript.

Figure 2

-The used buffer is described 0.04M BRB in the main text, but 0.01M BRB in Figure caption. The authors should check this condition. The other figure parts should be also checked by the authors.

-The authors should uniform the color of each plot in Figure 2a,b.

-The order of the current (Y-axis) between Figure 2a (10^-5 A order) and 2b(10^-6 A order) is quite different. I suggest that Figure 2a would be 10^-6 A order. Indeed, the author described “The bare GCE, pristine NiO and sample 1 did not show any electrochemical activity towards the oxidation of MTX.” (page 4, line 151; Figure 2b). If these are correct, these data show the similar results to Figure 2a. But both data were quite different currents even with the same CV shapes.

-According to the above comment, both data of S1 were quite different. In Figure 2a, background currents were gradually increased with (maybe) surface porosity (bare<pristine<S1<S2). On the other hand, Figure 2b showed the order S2<bare<pristine

Page 5, line 180

The term “Figure 4d” would be Figure 3d.

Figure 3

-It is unclear the tables in Figure 3a and b (also Figure 4b). I don’t think these are necessary.

-What is “IP” in the X-axis?

Figure 4

LSV and CV are the similar techniques, but the obtained performance were different as summarized in Table 3. The authors should address this concern.

Table 1

The notation “Signal increase (%)” is a little difficult to understand because the concentration of MTX and the others are 1/10 ratio. Is it possible to reconsider changing the notation here to clearly understand for the readers?

Table 3

The unit of LOD value should be unified to nM

Author Response

Reviewer 2

We are thankful to the reviewer for useful comments and suggestions prior to publication

Comments and Suggestions for Authors

This paper describes about NiO nanostructures for detecting methotrexate (MTX). I do believe this material is to be of interest and important. However, I don’t think this paper demonstrate the sufficiently scientific evidences and discussion to support the advantages of the developed MATERIAL. So I fail to see the quality and advantage of this study. Therefore, it is recommended that the authors should reconsider this paper (major revision).

Some comments are listed below.

Title

The title is included “new protocol for monitoring”. The proposed protocols are traditional electrochemical techniques (CV, LSV, and i-t). The authors should revise this part.

Ans. We have changed the title in the revised version of manuscript

The developed NiO nanostructures

I understand that the obtained sample 2 exhibited superior electroanalytical performance to pristine NiO and sample 1. But it is unclear this reason. Due to the size of porous structure (surface roughness, or porosity)? The authors should explain this reason sufficiently because this point is the most important in this manuscript.

Ans. The superior performance of sample is attributed to the several aspects of nanostructured material like the particle size, porosity of material and the surface roughness as they are providing the large number of catalytic sites for the oxidation of anticancer drug. The surface roughness and the porosity provide the exposure of large surface area for the catalytic reaction, thus superior performance is shown by the sample 2 in the reported work.

Figure 2

-The used buffer is described 0.04M BRB in the main text, but 0.01M BRB in Figure caption. The authors should check this condition. The other figure parts should be also checked by the authors.

Ans. This has been rectified throughout the manuscript

-The authors should uniform the color of each plot in Figure 2a,b.

Ans.; The suggested changes are made in the figure 2a,b.

-The order of the current (Y-axis) between Figure 2a (10^-5 A order) and 2b(10^-6 A order) is quite different. I suggest that Figure 2a would be 10^-6 A order. Indeed, the author described “The bare GCE, pristine NiO and sample 1 did not show any electrochemical activity towards the oxidation of MTX.” (page 4, line 151; Figure 2b). If these are correct, these data show the similar results to Figure 2a. But both data were quite different currents even with the same CV shapes.

Ans. We apologize for the inconvenience of the figure in the first submission. The Figure 2a and 2b have different volume of solution in the electrochemical cell as Figure 2a indicates the response of each material in the electrolytic conditions whereas the figure 2b shows the response of each material with electrolyte and the MTX as analytie, therefore this might influence on the value of current and we do see the similar CV features but with different current background.

-According to the above comment, both data of S1 were quite different. In Figure 2a, background currents were gradually increased with (maybe) surface porosity (bare<pristine<S1<S2). On the other hand, Figure 2b showed the order S2<bare<pristine

Ans. The Figure 2a and 2b have different volume of solution in the electrochemical cell as Figure 2a indicates the response of each material in the electrolytic conditions whereas the figure 2b shows the response of each material with electrolyte and the MTX as analyte, therefore this might influence on the value of current and we do see the similar CV features but with different background of current for the S1.

Page 5, line 180

The term “Figure 4d” would be Figure 3d.

Ans. This has been corrected in the revised version of manuscript.

Figure 3

-It is unclear the tables in Figure 3a and b (also Figure 4b). I don’t think these are necessary.

Ans. These changes are made in the revised version of manuscript.

-What is “IP” in the X-axis?

Ans.  We thank you to the reviewer for this correction. This has been corrected in the revised version of manuscript.  

Figure 4

LSV and CV are the similar techniques, but the obtained performance were different as summarized in Table 3. The authors should address this concern.

Ans. The CV and LSV have similar aspects but the LSV is half CV whereas the CV is accompanied by the both forward and reverse reactions and it might influence on the different performance of material given by LSV and CV. As CV uses the both forward and reverse process that may change the electrochemical properties of NiO. Also, we did not optimize the electrochemical cell conditions such as the optimum electrolyte volume, the position of working electrode inside the cell, and the catalytic material on the glassy carbon electrode. All these conditions should be controlled during the analysis of practical analysis of drugs or any other analyte. However, we just investigated the electrochemical properties of NiO for MTX and we report the best result that we obtained.  These ambiguities are detailed in the revised version of manuscript.

Table 1

The notation “Signal increase (%)” is a little difficult to understand because the concentration of MTX and the others are 1/10 ratio. Is it possible to reconsider changing the notation here to clearly understand for the readers?

Ans. These corrections are made in the Table 1. We have tried to make it clear during the revision for the better understanding of readers.

Table 3

The unit of LOD value should be unified to nM

Ans. Thank you and we have unified the unit of LOD into nM in the Table 3.

Round 2

Reviewer 2 Report

The resubmitted manuscript was well revised in accordance with the comments. Therefore, this paper is worthy for publication of Nanomaterials.